# Fast Detection of Different Water Contaminants by Raman Spectroscopy and Surface-Enhanced Raman Spectroscopy

**DOI:** 10.3390/s22218338

**Published:** 2022-10-30

**Authors:** Salvatore Almaviva, Florinda Artuso, Isabella Giardina, Antonia Lai, Alessandra Pasquo

**Affiliations:** ENEA, Italian National Agency for New Technologies, Energy and Sustainable Economic Development, Frascati Research Center, Via Enrico Fermi 45, I-00040 Frascati, Italy

**Keywords:** Raman spectroscopy, SERS, water pollution, PAHs, *Escherichia coli*, pesticides

## Abstract

Fast monitoring of water quality is a fundamental part of environmental management and protection, in particular, the possibility of qualitatively and quantitatively determining its contamination at levels that are dangerous for human health, fauna and flora. Among the techniques currently available, Raman spectroscopy and its variant, Surface-Enhanced Raman Spectroscopy (SERS), have several advantages, including no need for sample preparation, quick and easy operation and the ability to operate on the field. This article describes the application of the Raman and SERS technique to liquid samples contaminated with different classes of substances, including nitrates, phosphates, pesticides and their metabolites. The technique was also used for the detection of the air pollutant polycyclic aromatic hydrocarbons and, in particular, benzo(a)pyrene, considered as a reference for the carcinogenicity of the whole class of these compounds. To pre-concentrate the analytes, we applied a methodology based on the well-known coffee-ring effect, which ensures preconcentration of the analytes without any pretreatment of the sample, providing a versatile approach for fast and in-situ detection of water pollutants. The obtained results allowed us to reveal these analytes at low concentrations, close to or lower than their regulatory limits.

## 1. Introduction

Polluted waters represent a serious risk to human health [1] and the integrity of the ecosystem [2], affecting animals and plants and compromising them because they constitute a dangerous vehicle for toxic substances, viruses and bacteria, which can reach humans and cause the onset of a number of diseases, including serious ones [1,3]. It has been demonstrated that pollutants penetrate the food chain, causing diseases, malformations and the risk of loss of numerous species. Often pollutants are released in hot waters discharged from industrial plants, and this represents an element of further imbalance for many aquatic habitats [4] since they change their temperature and reduce the amount of available oxygen [5].

For the abovementioned reasons, fast environmental monitoring of waters is an important part of controlling pollutants and any effects they have on the health of humans and animals, both domestic and wild and on plants and vegetables.

Among the current detection techniques, it is worth citing optical and mass spectroscopy [6] techniques such as atomic fluorescence spectrometry [7,8], liquid chromatography [9,10], UV-VIS spectrophotometry [11], vibrational spectrometry [12], inductively coupled plasma mass spectrometry [13], graphite furnace atomic absorption spectrometry [14] and flame atomic absorption spectrometry [15]. However, these methods generally require a long detection cycle, and some of them need the use of specific chemicals which may cause secondary pollution.

Raman spectroscopy (RS) [16,17] is a valuable complementary technique to the previously described as it offers the distinct advantages of chemical specificity and benefits from the ability to generate valid reference spectra with less or no sample alteration. 

The Raman effect is a two-photon inelastic light-scattering event, where incident photons (typically from a laser source) are of much greater energy than the vibrational energy levels [16,17] of the molecules on which they impinge. The photons lose part of their energy to the molecular vibrations, with the remaining energy scattered as photons of reduced frequency. The frequencies of these molecular vibrations depend on the masses of the atoms of the substance analyzed, on their geometric arrangement and on the strength of their chemical bonds. Since the vibrational energy levels are unique to each molecule, the Raman spectrum provides a “fingerprint” of that molecule. RS is generally non-destructive, fast (typically few minutes or less are required to acquire a spectrum) and can be used for the in-situ analysis of tablets, powders, and liquids. This is particularly important with regard to the prevention of sample contamination and preservation for further analyses. 

Until a few years ago, Raman devices were restricted to laboratories due to their complexity in detecting the weak Raman scattering process. However, recent advances have overcome this drawback with the production of compact and field-portable Raman systems [18]. This technological advancement is due to the advent of compact, powerful, stable and reliable near-infrared solid-state laser sources and high-resolution charge-coupled device (CCD) detectors. Further developments concern software of spectral identification, simplifying the research of spectral structures that characterizes analytes of interest.

Surface-Enhanced Raman Spectroscopy (SERS) is a variant of RS where the sample is analyzed over a roughened noble metal surface or mixed with metal nanoparticles, and the Raman signal results are amplified up to several orders of magnitude [19]. In SERS, this amplification comes mainly through the electromagnetic interaction of light with metals [19], which amplifies the electric field carried by the laser photons through excitations of the electron of the metals, known as plasmon resonances [19]. The result is an intense electric field in which molecules under analysis are excited. To have the highest amplification, molecules must typically be adsorbed on the metal surface or at least be very close to it (typically 10 nm maximum) [19]. In this case, SERS has been proven to obtain a spectral signal even from analytes dissolved at extremely low concentrations, up to a few ppb or less [19]. 

Here we present our results of the application of RS and SERS for the detection of several pollutants, dissolved or suspended in drinking water, simulating pollution of its sources like rivers, lakes, and artificial basins, used not only for humans but also for animal consumption and for agriculture, representative of the common pollution processes induced by human activities. 

The pollution processes considered for Raman-SERS analyzes are the following:(1)eutrophication of surface waters for agricultural and urban runoff and industrial sewage and waste;(2)air (and water) pollution as a result of incomplete combustion or pyrolysis of organic material;(3)Presence of microorganisms, in particular harmful bacteria such as the Enterobacteriaceae, Enterococcaceae, Streptococcaceae and Vibrionaceae families.

Further details on the specific substances and microorganisms that have been considered are reported in Section 2 (Materials and Methods).

To increase the sensitivity of both RS and SERS to many analytes dissolved in water, a pre-concentration procedure is required, and we adopted the procedure suggested by Xu et al. [20] based on the so-called “coffee-ring” effect [21].

These substances were detected in concentrations of a few mg/L or less, in some cases lower than the limit for water potability established by World Health Organization (WHO). In the case of suspended substances, we identified their characteristic spectral signature on solid traces of few ng.

## 2. Materials and Methods

Raman-SERS measurements were performed with a table-top micro-Raman system “i-Raman” by Bw&Tek, equipped with a Gallium Aluminum Arsenide (GaAlAs) diode laser emitting at 785 nm (linewidth < 0.3 nm). The investigated spectral range was between 175–3250 cm^−1^ (796–1054 nm) with a spectral resolution of 4.5 cm^−1^. 

The optical geometry of the device is confocal; the laser light and the Raman signal are focused and collected through an optical microscope equipped with different objectives: 10X, with a numerical aperture (N.A.) 0.12 and a laser spot of 180 μm diameter, 20X, N.A. 0.4 and a laser spot of 90 μm diameter, 40X, N.A. 0.5 and a laser spot 40 μm diameter, 80X, N.A. 0.75 and a laser spot of 25 μm diameter. Measurements were performed by using the 40X objective, representing the right compromise between a relatively large scanned area that limits the fluctuations of the Raman-SERS signal in different points of the sample and good collection efficiency, increasing SNR (Signal-to-Noise Ratio). The signal was spectrally analyzed through a 600 lines/mm grating and recorded by a cooled CCD (10 °C) array detector (2048 pixels, 16 digital bits). Each spectrum was the average of 3 subsequent acquisitions. Laser power was adjusted from a few tenths up to 300 mW, depending on the Raman cross-section of the sample. Before each series of measurements, the instrument was preliminarily corrected for the spectral contribution of the environment by acquiring and subtracting a background signal obtained without laser excitation. Then a spectral calibration was performed by measuring a reference sample of polytetrafluoroethylene (PTFE), whose spectral structures are well-known in the literature. 

The protocol of measurements involved scanning the ring-shaped residue formed by the substance in a drop of water (coffee ring) after its evaporation [20]. In this way, the substances are concentrated without any complex and potentially polluting preparation and can be easily measured, also in an in-situ campaign. To avoid the uneven distribution of the analyte in the coffee ring, we repeated the measurements at different points of the “coffee ring” for each sample, generally close to the very edge of the deposit, where the signal was higher, and we checked for the repeatability of the spectral signal.

The volume of the drops was between 2 and 5 μL, with an evaporation time of 15–20 min (RT ~ 20°) and a residual coffee ring of 1–2 mm diameter. In the case of Raman measurements, the drops were deposited on standard microscope glasses, covered with an aluminum (Al) foil, because metals do not introduce any interfering features in the Raman spectra [16]. In the case of SERS measurements, the drops were deposited in two different ways: (1) on planar, nanostructured SERS substrates or (2) added with an aliquot of suspended gold nanoparticles (g-NPS), then deposited on the Al-covered microscope glass slide. 

We used common drinking water but also de-ionized (DI) water as a solvent to highlight any interference in the spectrum due to the presence of ions already dissolved in drinking water and to evaluate whether, with the adopted procedure, the sensitivity of the technique is significantly compromised.

The water pollution processes considered for the analyses are the following:(1)The eutrophication of surface waters due to agricultural and urban runoff and sewage and industrial waste. This can lead to harmful algal blooms due to the availability of nutrients such as nitrogen, phosphorus or sulfur [22,23]. Massive and prolonged algal bloom leads to hypoxia, resulting in aquatic life die-off or large pollution of water by microcystins. Contamination of drinking water sources requires a multiple-barrier approach in removing diverse contaminants, either natural and/or anthropogenic, which is often expensive;(2)Air (and water) pollution as a result of incomplete combustion or pyrolysis of organic material [24,25];(3)The presence of microorganisms, in particular harmful bacterial species like Entero-bacteriaceae, Enterococcaceae, Streptococcaceae, and Vibrionaceae families. The presence of pathogenic microorganisms in groundwater is important because, due to the recent prolonged periods of drought and the pressing demand for water for crops, the Food and Agriculture Organization of the United Nations (FAO) estimates that 70% of water consumption is designated to agriculture, so there is a general guideline to use wastewater for irrigation. Despite the fact that this approach would help to preserve the freshwater sources and recycle the nutrients present in wastewater, it would pose the problem of the presence of harmful pathogens on fruits and vegetables, causing severe diseases in humans [26];(4)The massive use of pesticides. They are nowadays widely used in agriculture to prevent or eliminate the action of insects on agricultural products [27,28]. However, some studies show that pesticides may have negative effects on consumers [29,30,31], so a fast detection of pesticide residues is of increasing importance because their intensive use and their persistence made them widespread in the environment, and some traces have also been revealed in the aquifers destined for human consumption.

In particular, the substances that have been investigated are as follows:

(a) Nitrates, phosphates and sulfites: they are the products of anthropogenic activities in the industrial and agro-food fields. They act as nutrients in the aquatic environment generating anomalous algal bloom. High concentrations of nitrates in drinking water present serious risks for humans, causing methemoglobinemia in children (blue baby syndrome) [32] through their conversion to nitrites (under anaerobic conditions in the intestine) and subsequent blocking of the oxygen-carrying capacity of hemoglobin [33]. Furthermore, ingested nitrates have a potential role in the development of cancers of the digestive tract through their contribution to the formation of nitrosamines, which are among the most potent carcinogens known in mammals. The Italian law sets a limit of 50 mg/L for nitrate as an ion in drinking water [34] which is the same established by WHO [35].

The contamination of water by ammonium nitrate was simulated by preparing solutions at a known concentration of NH_4_NO_3_ (Sigma, purity > 98%) both in DI and drinking water (drawn from the southern Rome area) and finally adding an aliquot of g-NPS with a concentration of about 5 × 10^−4^ mg/mL.

Phosphates, their presence in inland waters comes from different sources like polyphosphates from domestic waste containing detergents, orthophosphates from the run-off waters of the land treated with fertilizers, organic phosphates from pesticides, phosphates entering through plants and rocks or intentionally dissolved in water to prevent lead poisoning or metal dissolution of water of pipes. Phosphates are believed to be safe to ingest, they are important complexed components of all plant and animal-based foods, but their excess in water causes eutrophication, excessive algal growth and consequent reduction of oxygen. 

The contamination of water by orthophosphate ions was simulated by preparing solutions of potassium hydrogen phosphate (KH_2_PO_4,_ Sigma, purity > 99%) at concentrations from 1 to 100 mg/mL in drinking water. 

Sulfites are preservatives used for food conservation to avoid the formation of potentially harmful microorganisms and delay degradation processes, having antioxidant, antibiotic and antiseptic purposes. In wine production, these substances are added in different stages and with different purposes because sulfitation prevents the oxidation of grape juice and inhibits fermentation activated by yeasts present on the skin of the berries, which could negatively affect the final aromas of the wine. Albeit in small quantities, they are naturally present in wines up to 30 mg/L. In white wines and sweet wines instead, the artificial addition of sulfites can reach concentrations up to 400 mg/L. Sulfite ions also play crucial roles in the atmospheric environment [36,37], leading to the formation of acid rain after reacting with water [38]. The excessive intake of this anion is poisonous and may cause lung and brain cancer, strokes, migraine headaches, asthma attacks, and myocardial ischemia [39,40,41]. Sodium sulfite (Na_2_SO_3_, Sigma, purity > 98%), used in the textile industry as a whitening, desulphurizing and dechlorinating agent, for example, in swimming pools, was considered as representative of the whole class of sulfites.

The contamination of water was simulated by preparing a solution of Na_2_SO_3_, 500 mg/L in drinking water, the maximum level suggested by WHO in the Guidelines for Drinking-water Quality [35]. 

(b) Polycyclic aromatic hydrocarbons (PAHs) are present in the air as a result of incomplete combustion or pyrolysis of organic material, such as coal, wood, petroleum products and waste [42]. PAHs are ubiquitous in the environment as they can be found in air, soil, sediments, water and also in food. Some PAHs have been classified as human carcinogens by the International Agency for Research on Cancer [43]. The published research about the toxic effects of PAHs on living organisms has led the main international agencies of environmental protection to include them in the list of priority substances of special concern. Sixteen PAHs have been recognized by U.S. Environmental Protection Agency (EPA) as priority pollutants. PAHs can reach water bodies mainly through dry and wet deposition, road runoff, industrial wastewater, petroleum spills, and fossil fuel combustion [44,45,46,47]. Due to their stability and poor solubility in water may bio-accumulate in aquatic species and have the potential for long-range transport [48].

The contamination of water by PAHs was simulated by preparing suspensions on anthracene (ANT) and Benzo[a]Pyrene (BaP) and, finally, a mix of several PAHs. The sample of ANT was prepared from a 1:1 mixture of a standard solution (Supelco, purity > 99%) of ANT in CH_3_OH:CH_2_Cl_2_ (0.2 mg/mL) and a suspension of g-NPS 0.1 mg/mL in DI. The total concentration of the solute was 0.05 mg/mL. Measurements were carried out by depositing a drop of 5 μL of the mixture after the solvent’s evaporation. The sampled quantity was estimated by observing at the optical microscope that the coffee ring had a diameter of about 2 mm and a width of about 200 μm. Assuming that all the compound dissolved in the drop (2.5∙10^−3^ mg) was uniformly deposited in the coffee ring, a scanned quantity of ≈ 35 ng of sampled analyte was estimated by taking into account a scanned area of 40 μm diameter (laser spot diameter). BAP is primarily found in cigarette smoke, in gasoline and diesel exhaust, and in the fumes produced by the combustion of biomasses, charcoal-broiled foods, petroleum asphalt and shale oils. BaP is the most widely studied PAH from a toxicological point of view and is most frequently determined in the environmental and food matrices. It is frequently used as an indicator of the class of PAHs, as regards both the levels of contamination and the cancer-causing risk. The BAP sample was prepared from a solution of BaP (Supelco, purity > 98.6%) in CH_3_OH:CH_2_Cl_2_ 1:1, mixed with g-NPS at a concentration of 0.2 mg/mL and the measurements were carried out by depositing a drop of 5 μL of this suspension.

(c) *Escherichia coli* (EC) is used as an indicator of human pollution since it is present in the intestine of warm-blooded animals. Its presence in inland waters has an impact on human health, leading to many skin and intestinal diseases, [49] making it necessary to monitor it carefully. Two different strains of EC were used in this study: DH5α and BL21-DE3. Both were purchased by ATCC (ID ATCC 12 435) and are all Lamba-derivative of EC laboratory strain K-12. Bacterial strains were grown in 10 mL of Luria- Bertani broth (Sigma) at 37 °C for a period of time to reach the log phase, then harvested by centrifugation. The log phase was determined by plotting the logOD at 600 nm (with a Jasco Global UV-Visible Spectrophotometer) vs. time. For each strain used in our experiments, the mid-log phase corresponded between 0.5 and 0.9 AU. Finally, bacteria were washed exhaustively with DI. The resulting pellet was re-suspended in 250 μL of DI in order to get rid of the growth medium that could leave precipitants upon coffee-ring formation once the solvent had evaporated. The cell count was determined with the direct microscopic counting methods using a hemocytometer, resulting in approximately 10^9^ cell/mL bacterial suspension. This starting suspension originated serial dilution samples of one order of magnitude each (10^8^, 10^7^, 10^6^, 10^5^ and 10^4^ cell/mL bacterial suspensions, respectively). A drop ranging from 3 to 5 μL of each suspension was pipetted directly onto the SERS substrate for the purposes of the data analysis described here.

(d) Glyphosate, or N-(phosphono-methyl) glycine, is probably the most widely used pesticide in the world [27]. It acts as an enzyme inhibitor of plants, thus eliminating weeds and increasing the productivity of the field [28]. Glyphosate is a broad-spectrum, non-selective herbicide, water-soluble and chemically stable. Its main metabolite, Aminomethylphosphonic acid (AMPA), is produced by soil bacteria [29]. Both are persistent in the environment and can be found in soil, air, water, as well as groundwater and food products. Although glyphosate is a widely used herbicide employed, recently, the dangerousness hazard and toxicity of glyphosate and AMPA have been demonstrated for human health [29,30,31,49] and the environment.

In 2015 the WHO’s International Agency for Research on Cancer (IARC) classified glyphosate as “probably carcinogenic in humans” [50]. Even if the toxicity of AMPA was reported to be less than or comparable to glyphosate [51], due to its protracted persistence in the environment, several studies established that environmental AMPA harmfulness toxicity was higher than that of glyphosate [52,53].

Despite their limited capability to penetrate the soil to about 20 cm [54] and the probability that they can reach the groundwater, many studies indicate the presence of both in aquifers [54,55] due to their high solubility. Moreover, due to the complexity of the available techniques for their detection and the nature of the molecules, glyphosate and AMPA are not normally mandatory in routine sampling monitoring. Thus, a simple method that allows us to perform measurements without or with the negligible use of reagents represents a huge advantage. Both pesticides were measured after dilution in drinking water at a concentration of 1.54 × 10^−2^ M (2.6 mg/mL) for Glyphosate and 6.5 × 10^−5^ M (7.2 mg/L ≈ 11.2 ppm) for AMPA by depositing a drop of 5 μL of suspension on the SERS substrates MATO-S™ by Integrated Optics. 

All the solutions/suspensions have been prepared and measured on the same day or stored at 4 °C and measured the following day. 

## 3. Results and Discussion

### 3.1. Nitrates

The nitrate ion has a Raman spectrum showing the characteristic peak at 1045–1049 cm^−1^, relative to the ν1 symmetric stretching of this molecular group [56], which is used as a spectral reference structure for the detection of the ion [56]. In Figure 1, the Raman spectrum of a crystal of ammonium nitrate (NH_4_NO_3_) shows such a spectral feature:

The nitrate ion was in DI water at 1 mg/L in DI. A drop of 1 μL of this solution was deposited on the Al foil. Part of the coffee ring and the relative SERS spectrum are shown in Figure 2a,b:

In drinking water, the characteristic NO_3_^−^ peak was still detected in a solution with a minimum concentration of 20 mg/L NH_4_NO_3_, which is still within the limits allowed by the Italian law DL n. 31 2Febbraio 2001 (dlgs-31-2001) [34] and WHO regulation concerning the nitrate ions concentration dissolved in water [35]. The corresponding SERS spectrum is reported in Figure 3. As reported in Isman et al. [57], the presence of other anions (the distributor’s data report in particular bicarbonates, HCO_3_^−^ 405 mg/L and sulfates SO_4_^2−^ 17.2 mg/L) reduces the available SERS “hot spots,” i.e., the highest SERS amplificationpoints [57] for the NO3- ions and the overall sensitivity of the technique for any specific ion [57]. 

### 3.2. Phosphates

The Raman spectrum of KH_2_PO_4_ in crystalline form is shown in Figure 4. The spectral reference structure for the detection of the anion PO_4_^3−^ is at about 915 cm^−1^, assigned to the ν1 (νP–OH) vibration mode of the ion [58,59].

In good accordance with literature data [57], PO_4_^3−^ ion was detected with a lower sensitivity if compared with the nitrate one. Our method managed to detect a peak at 915 cm^−1^, corresponding to the spectral features of the PO_4_^3−^ anion, only at a concentration of 1 mg/mL and only with an increased acquisition time of up to 120 s and a laser power of up to 300 mW. The resulting Raman spectrum is shown in Figure 5. The detected concentration of the PO_4_^3−^ anion with our method is well above the recommended maximum contaminant level (MCL). Surface waters have PO_4_^3−^ anion concentration limits of 0.1 mg/mL, which is the MCL acceptable for the prevention of eutrophication. Furthermore, reservoir water and drinking water have an even lower MCL of 0.05 mg/mL and 0.025 ng/mL, respectively, to meet the criterion of uncontaminated water. For this reason, we acknowledge that our protocol does not meet the required sensitivity for detecting PO_4_^3−^ anion concentrations close to the limits suggested by international and national organizations such as WHO.

### 3.3. Sulphites

The Raman spectrum of Na_2_SO_3_ in its crystalline form is shown in Figure 6. If sodium sulfite is allowed to crystallize from an aqueous solution at room temperature, it generates heptahydrate, [60] oxidizing in the air to form sulfates (SO_4_^2−^) ions, so it is expected to detect some spectral features relative to sulfates [60]. 

The spectrum shows the most intense bands at 950–988 cm^−1^, assigned to the ν_1_ vibration of the SO_4_^2−^ anion, whereas the bands at 498 and 640 cm^−1^ are ascribed to the ν_2_, ν_4_ vibrations, respectively [57]. The ν_1_ vibrational band centered at 988 cm^−1^ is specific to the sulfate anion and was observed at a concentration of 500 mg/L after integration of 120 s, as shown in Figure 7. This result indicates that sulfates can be detected in drinking water at concentrations within the suggested WHO limit [35,61]. 

### 3.4. Polycyclic Aromatic Hydrocarbons 

#### 3.4.1. Anthracene

Although its carcinogenicity has not been confirmed by the bioassays, ANT is on the EPA’s priority pollutant list and was recently listed by the European Chemicals Agency (ECHA) among the substances of very high concern (SVHC) [61,62,63,64] because it is persistent and can bioaccumulate and it is toxic to aquatic organisms. 

As shown in Figure 8, the spectral signatures of anthracene from a solution of 50 mg/L were evidently detected at about 394 cm^−1^, 751 cm^−1^ and1400 cm^−1^ in agreement with literature data [63].

#### 3.4.2. Benzo[a]Pyrene (BaP)

The Raman-SERS spectrum of BaP is shown in Figure 9 and was obtained by depositing the sample on nanostructured SERS substrates (MATO-S™ by Integrated Optics). As in the case of ANT, the sample was measured as a solid residue of a solution of BaP in CH3OH:CH2Cl2 mixed with g-NPS at a final concentration of 7.9∙10-4 M (200 mg/L). The scanned material was estimated by assuming that the whole substance dissolved in the 5 μL drop (1 μg) was homogeneously deposited in the coffee ring, whose width was estimated to be ~ 25 μm (inset of Figure 9) and diameter was ~ 2 mm. With a laser spot of 40 μm diameter, the quantity of sampled compound was estimated to be ~ 3.25 ng.

#### 3.4.3. PAHs Mix

In incomplete combustion of modern biomass (such as wood) and fossil fuels (petroleum and coal), many PAHs are simultaneously formed. Therefore, a more realistic scenario of PAHs detection includes the analysis of a mixture of these substances. The procedure applied to ANT and BaP has been used with a standard solution of PAHs mix (EPA 610 mix by Sigma Aldrich) in methanol (MEOH), containing these compounds with concentration from 100 to 2000 μg/mL, as reported in the Table 1:

Table 1 summarizes the concentration of the components for EPA 610 mix (Sigma Aldrich).

As shown in Figure 10, in this complex mixture sample, the SERS spectrum exhibited several spectral signatures ascribable to single components of the mixture, which is in good agreement with the results reported in the literature [64]. 

### 3.5. Escherichia Coli

Figure 11 shows two clusters of EC: the first was deposited on a nanostructured SERS substrate (Klarite™ from Renishaw), and the second was mixed with g-NPS deposited onto the Al-covered microscope glass.

The analysis of EC suspensions in culture media did not provide any spectral signal at any of the prepared concentrations [65,66]. To overcome this problem, we applied a preconcentration step recommended by Hamasha et al. [67], which includes repeated washing and centrifugation of the bacterial pellet in ultrapure water. The pellet was then extracted and measured, allowing to obtain the Raman spectrum of the microorganism with an integration time of 60 sec and laser power of 240 mW, with spectral features emerging from the residual fluorescence in good agreement with literature data [67,68]. Figure 12 shows the spectrum obtained on the pellet of 10^9^ cell/mL bacterial suspension with the relative spectral assignments according to Kusic et al. [68].

### 3.6. Pesticides

#### 3.6.1. Glyphosate

The Raman spectrum of glyphosate in crystalline form is shown in Figure 13. The spectrum of the diluted substance (1.54 × 10^−4^ M, 2.6 × 10^3^ mg/L), also shown in Figure 13, presents very weak bands corresponding to those of the substance in the crystalline form [69].

#### 3.6.2. AMPA

The detection of AMPA in drinking water appeared to be more effective than glyphosate, as it was possible to identify the Raman structures of the analyte at concentrations down to 6.5 × 10^−5^ M (7.2 mg/L). This is shown in Figure 14, where the most intense and relevant spectral features of AMPA in crystalline form matched those of the diluted sample when compared.

Table 2 summarizes the main experimental results reported in the text (contaminant used, effectiveness of the procedure, quantity detected).

## 4. Conclusions

In this study, we report the Raman/SERS analysis of samples of DI and drinking water spiked with chemicals and micro-organisms representative of inland water pollution processes of anthropogenic sources. The coffee-ring method was used to carry out fast analyses, requiring a few μL of contaminated water- samples. It was applied to a spectrum of substances that are difficult to measure simultaneously with the same analytical technique. In the case of nitrate and sulfate ions, the methodology was effective in detecting them in concentrations within the limits established by various national and international entities (50 mg/L for the nitrate ion and 500 mg/L for the sulfate ion). In the case of the phosphate ion, the procedure was not effective in detecting the substance within regulatory limits.

PAHs, in particular, anthracene and the carcinogen benzo[a]pyrene were detected in quantities of ng on the solid residue of the drop, starting from a suspension in water at a concentration of 50 mg/L for anthracene and 200 mg/L for benzo[a]pyrene. The same procedure was applied to a mix of PAHs, where the spectral contributions of the single components were identified.

The coffee-ring analysis of water contaminated with *Escherichia coli* did not yield significant results. However, implementing the procedure as reported in [54], through repeated washing and centrifugation of the bacterial pellet in DI, allowed us to obtain the spectral signature of the bacterium from a 10^9^ cells/mL bacterial suspension, in accordance with literature data.

Finally, the “coffee-ring” analysis of pesticides dissolved in water allowed us to spectrally detect glyphosate and its metabolite AMPA, although at higher concentrations compared to the regulatory limits for these substances.

In conclusion, the analysis of the “coffee-ring” residue has the advantage of being simple and fast and appears to be promising as it has been shown to obtain the spectral signature of specific contaminating analytes, even in the case of common drinking water samples containing interfering substances. However, the obtained results also show that the sensitivity of the technique is still quite low, and the detection limit is far from those of more complex standard techniques. These limits can be reduced by optimizing the SERS technique for this specific use, for example, by testing additional active SERS substrates or by using suspensions of nanoparticles of different concentrations and shapes.

## Figures and Tables

**Figure 1 sensors-22-08338-f001:**
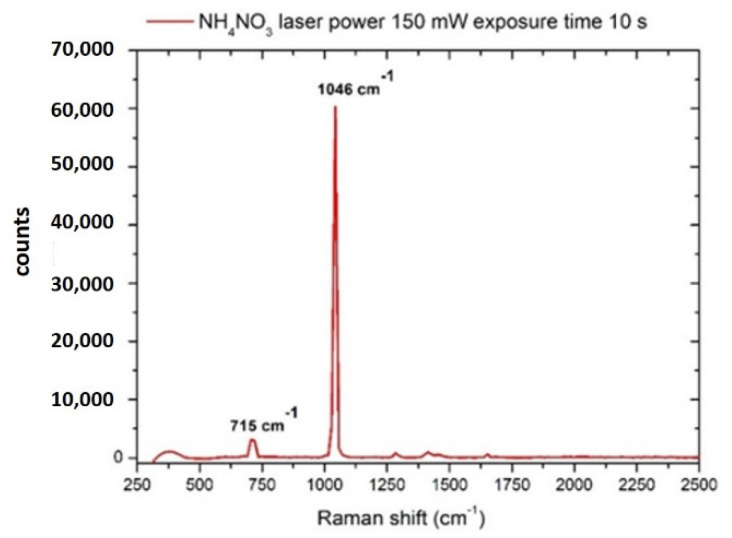
Raman spectrum of ammonium nitrate (NH_4_NO_3_) crystal. Laser spot 40 μm, laser power 150 mW, integration time 10 s.

**Figure 2 sensors-22-08338-f002:**
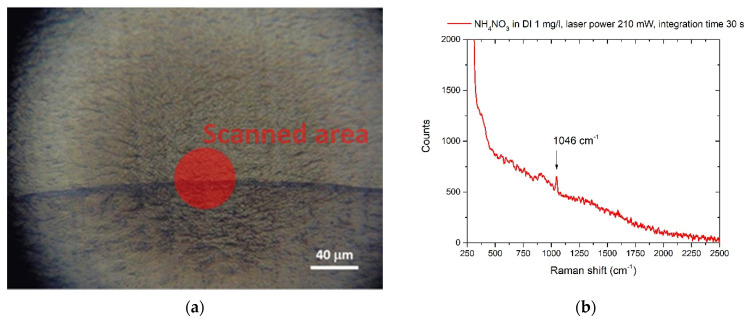
(**a**) coffee-ring section due to the residue of g-NPS and ammonium nitrate after the evaporation of 1 mg/L solution in DI. The red circle indicates the laser-scanned area with the 40X objective (**b**) Raman spectrum of ammonium nitrate dissolved in DI (1 mg/L), spot laser 40 μm, laser power 210 mW, integration time 30 s.

**Figure 3 sensors-22-08338-f003:**
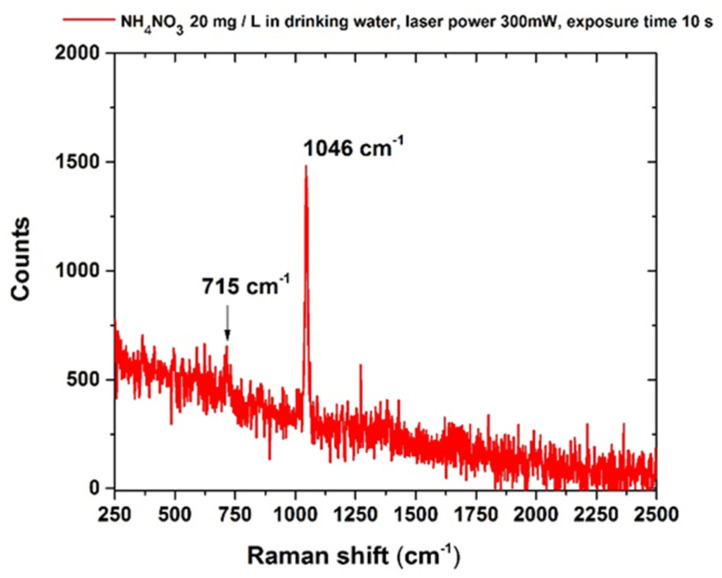
Raman spectrum of ammonium nitrate (NH_4_NO_3_) in drinking water (20 mg/L), acquired after the procedure described in Section 2. Laser spot 40 μm, laser power 300 mW, integration time 10 s.

**Figure 4 sensors-22-08338-f004:**
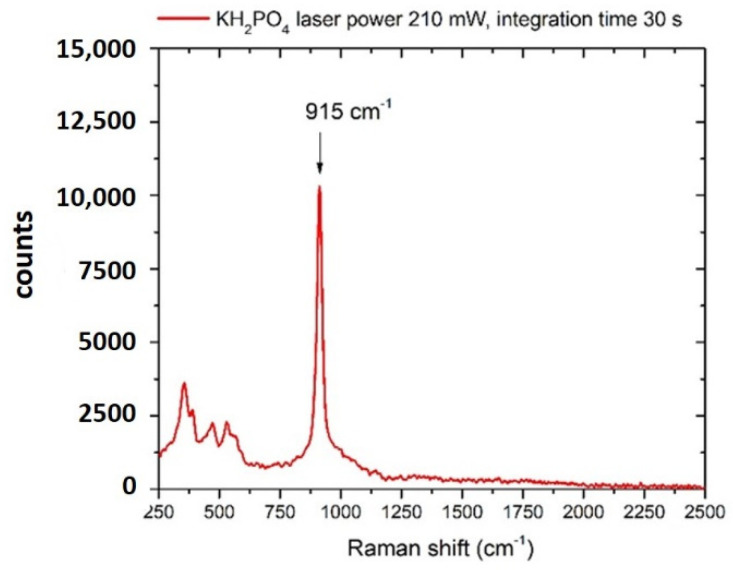
Raman spectrum of potassium hydrogen phosphate (KH2PO4) crystal. Laser spot 40 μm, laser power 210 mW, integration time 30 s.

**Figure 5 sensors-22-08338-f005:**
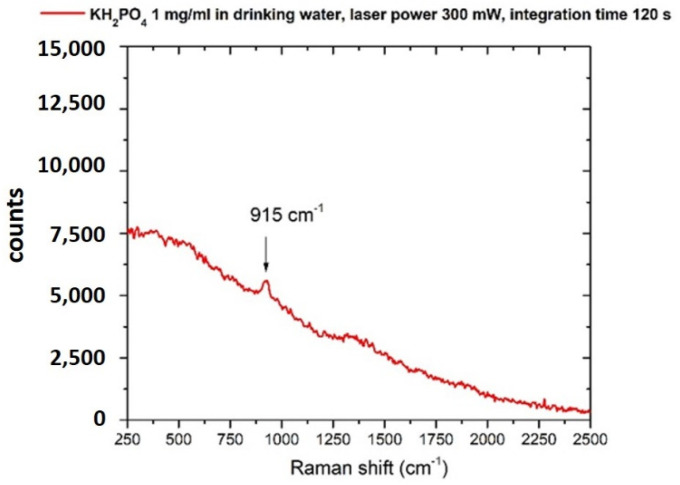
Raman spectrum of potassium hydrogen phosphate (KH_2_PO_4_) dissolved in drinking water (concentration 1 mg/mL). Laser spot 40 μm, laser power 300 mW, integration time 120 s.

**Figure 6 sensors-22-08338-f006:**
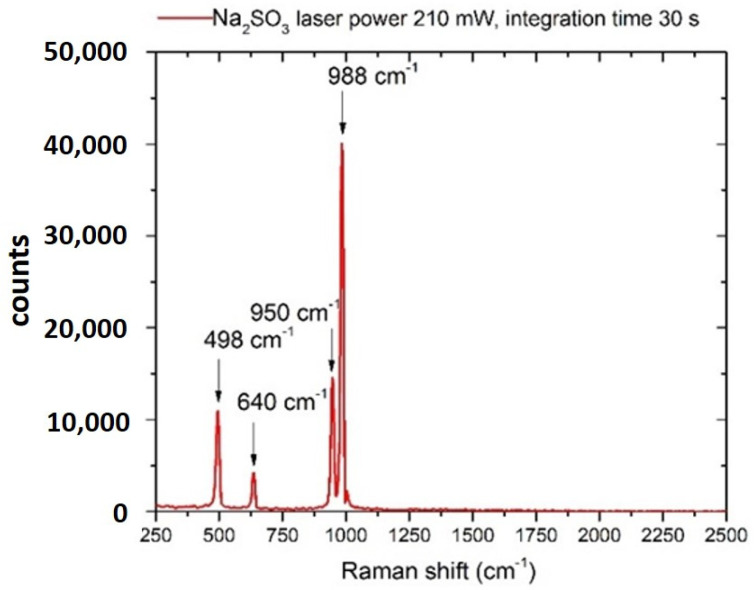
Raman spectrum of a sodium sulfite (Na_2_SO_3_) crystal. Laser spot 40 μm, laser power 210 mW, integration time 30 s.

**Figure 7 sensors-22-08338-f007:**
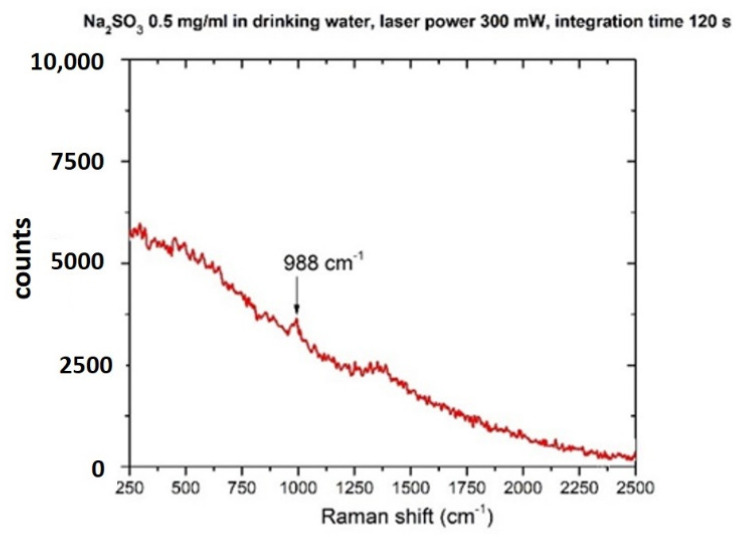
Raman spectrum of sodium sulfite (Na_2_SO_3_) dissolved in drinking water (concentration 500 mg/L). Laser spot 40 μm, laser power 300 mW, integration time 120 s.

**Figure 8 sensors-22-08338-f008:**
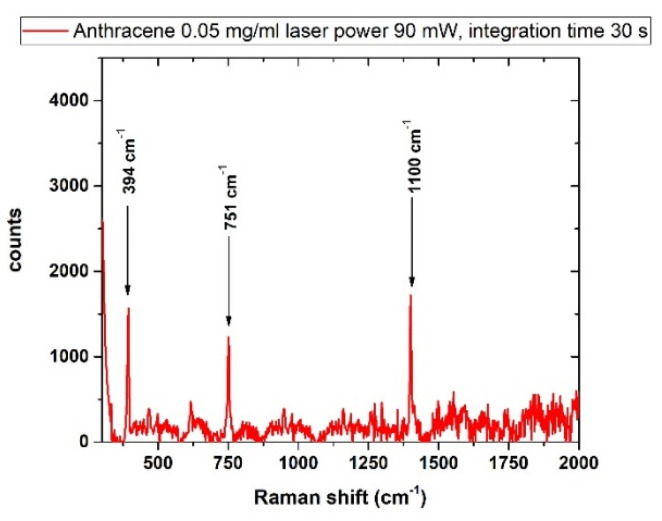
Raman spectrum of anthracene, laser power 90 mW, integration time 30 s. The spectral signatures of the substance are reported and found in agreement with the literature [53].

**Figure 9 sensors-22-08338-f009:**
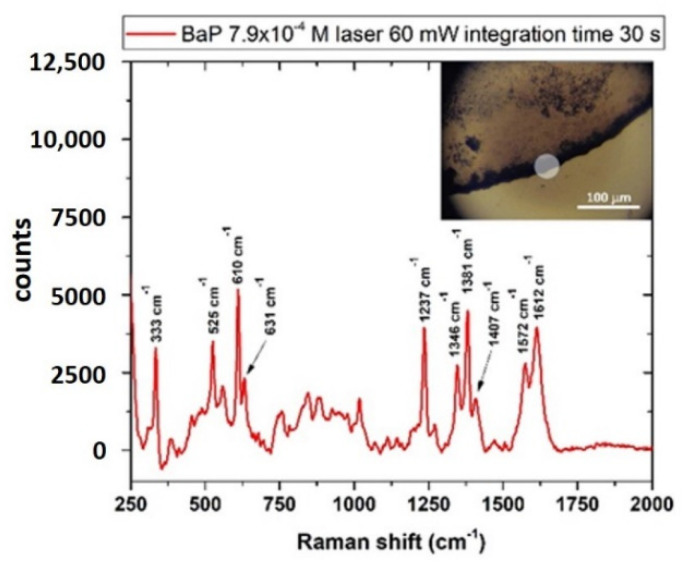
SERS spectrum of BaP with g-NPS on the substrate “Mato-s.” Laser power 60 mW, integration time 30 s. The inset shows a section of the coffee ring with some g-NPS clusters (dark areas) and a representation of the scanned area (white circle).

**Figure 10 sensors-22-08338-f010:**
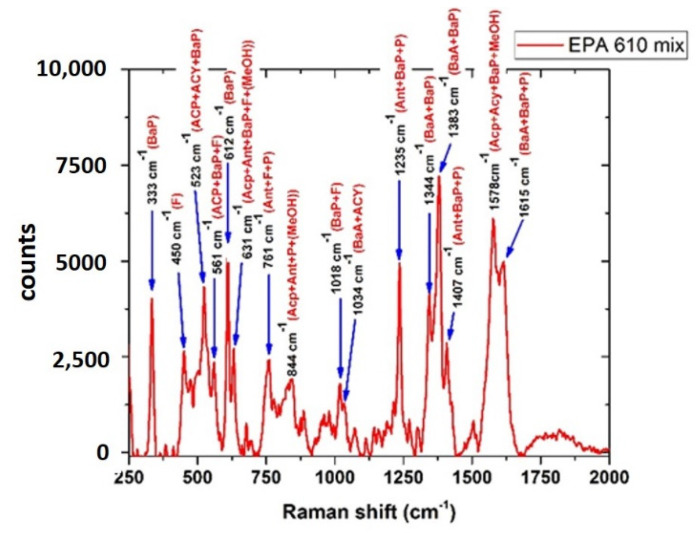
SERS spectrum of the EPA 610 mix and relative spectral signatures of the components. ACP = Acenaphthene, ACY = Acenaphthylene, ANT = Anthracene, BaA = Benz(a)anthracene, BaP = Benzo(a)pyrene, F = Fluorene, P = Pyrene, MeOH = Methanol (as residual traces of the solvent).

**Figure 11 sensors-22-08338-f011:**
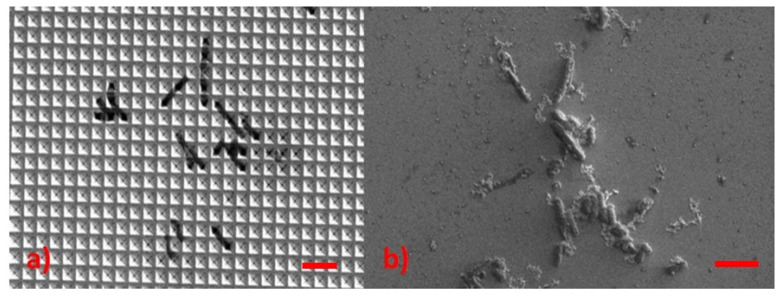
(**a**) Individual EC bacteria deposited on the SERS klarite substrate. (**b**) E. coli with g-NPS aggregates deposited on aluminum foil: reference bars (in red) are 5 μm long.

**Figure 12 sensors-22-08338-f012:**
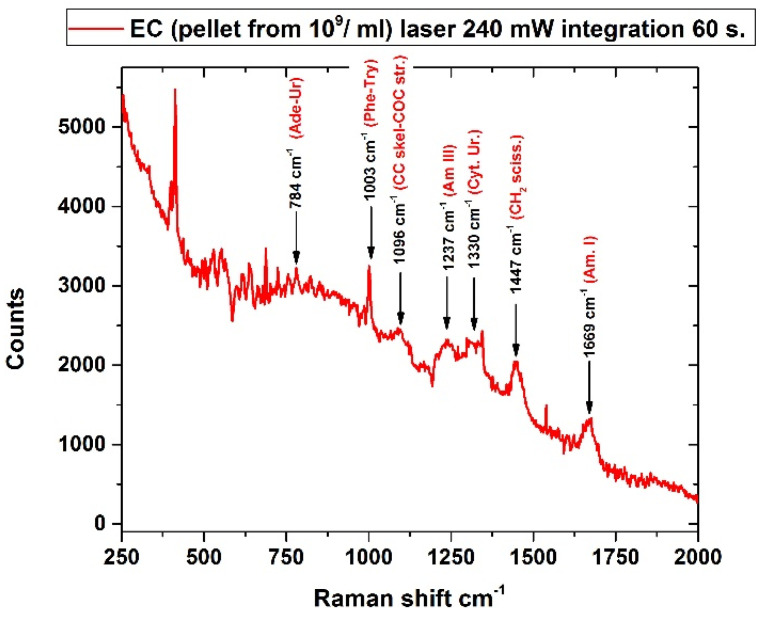
Raman spectrum of an EC (BL21 DE3) pellet extracted from a 10^9^/mL bacterial suspension. The spectral features ascribable to specific components according to literature data [68] are indicated with arrows. Legend: Ade = adenin, Am. I = amide I, Am. III = amide III, CC = C-C skeletal vibration, CH_2_ sciss.= CH_2_ scissoring vibration, COC str. = C-O-C stretching, Cyt. = citosine, Phe = phenilalanine, Try = tryptofan, Ur. = uracil.

**Figure 13 sensors-22-08338-f013:**
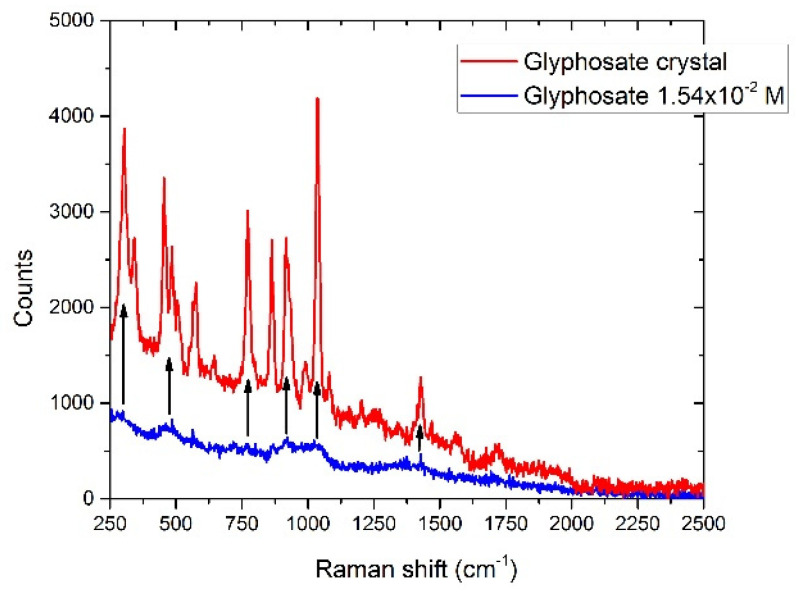
Raman spectrum of glyphosate crystal (red line) compared with the analyte dissolved in drinking water (blue line), concentration 1.54^−2^ M. The black arrows indicate the Raman bands of the crystal detectable in the diluted sample. Laser spot 40 μm, laser power 300 mW, integration time 120 s.

**Figure 14 sensors-22-08338-f014:**
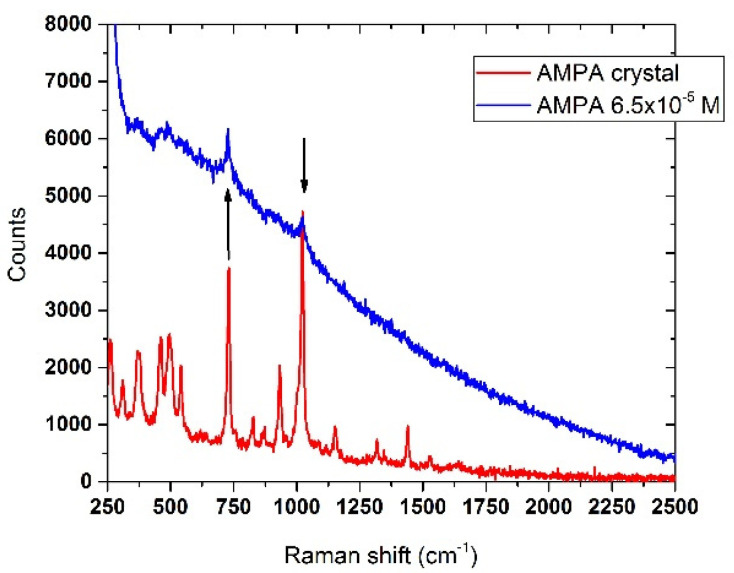
Raman spectrum of AMPA crystal (red line) compared with the analyte dissolved in drinking water (blue line), concentration 6.5 × 10^−5^ M (7.2 mg/mL). The black arrows indicate the Raman bands of the crystal detectable in the diluted sample. Laser spot 40 μm, laser power 90 mW, integration time 30 s.

**Table 1 sensors-22-08338-t001:** Composition of EPA 610 mix.

Description	Concentration (μg/mL)
Acenaphthene	1000
Acenaphthylene	2000
Anthracene	100
Benz(a)anthracene	100
Benzo(b)fluoranthene	200
Benzo(k)fluoranthene	100
Benzo(ghi)perylene	200
Benzo(a)pyrene	100
Chrysene	100
Dibenz[a,h]anthracene	200
Fluoranthene	200
Fluorene	200
Indenol [1,2.3-ca] pyrene	100
Naphthalene	1000
Phenanthrene	100
Pyrene	100

**Table 2 sensors-22-08338-t002:** Composition of EPA 610 mix.

Contaminant	Concentration (mg/L)
Nitrates	20
Phosphates	1000
Sulfites	500
Anthracene	50
Benzo[a]pyrene	200
PHAs mix	100–2000 *
Escherichia Coli **	10^9^ ***
Glyphosate	2.6 × 10^3^
AMPA	7.2

* Depending on the analyte in the mix. ** The preconcentration needed, according to Hamasha et al. [67]. *** Expressed in cells/mL.

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
