# Peer review of "Fast Detection of Different Water Contaminants by Raman Spectroscopy and Surface-Enhanced Raman Spectroscopy"

_sensors, 2022, doi:10.3390/s22218338_

Round 1

Reviewer 1 Report

In this study “Fast detection of different water contaminants by Raman spectroscopy and Surface-Enhanced Raman Spectroscopy”, the topic is interesting, and overall for the English language, minor spell checks are required in the paper. The manuscript should be edited to further improve the result discussion and the order of some figures must be checked. Here are my suggestions:

  1. A more literature review on the research status of pollutants and their effects is suggested in the introduction part. Especially from lines 24 to 37.  
  2. Please consider reviewing the abstract and the introduction to highlight the novelty and major findings.
  3. In line 30, please add “and” before plants and crops…animal, plants, and crops..
  4. In line 62 please consider using overcome instead of overcame
  5. In line 67 please use “has” instead of “have” (spectra identification software has..) or use the plural form.
  6. In the Results and discussion section, a description of the contaminants is given, this part should be included in the “Materials and Methods” section to describe and discuss the Raman and SERS spectra. As well as the description of the preparation of the simulated contaminants should be included in the materials and methods.
  7. Please specify the meaning of the acronyms the first time you cite them. In lines 122, 152, and 167, specify g-NPS, AMPA, and WHO. For example, AMPA is specified on page 14 (line 458) but you mention it before in the text as well as g-NPS which is clarified in line 328.
  8. In line 169, it is reported the sentence “the nitrate ion has a Raman spectrum showing the characteristic peak at 1045-1048 cm-1” which is supported by reference 28. In this reference the NO3 group is at 1049 cm-1, please consider extending the energy range to include such value.
  9. Add a few lines to explain why you used de-ionized (DI) water and drinking water in your experiments.
  10. In figure 2a, consider adding a circle and text to the image to explain to the readers what they are looking at.
  11.  Figure 3 is on page 5 and its caption in the next page. Please check it.
  12. In line 236 check the units cm-1 as well as in line 260.
  13. The description of Section 3.4 Polycyclic aromatic hydrocarbons, does not include relevant results of your work. I suggest resuming it and including it in the material and methods.
  14.  Please describe all the peaks in the Raman spectra in figure 5. If they are not representative of your work add a few lines to explain why.
  15. In all the figures, you used integer numbers for the peak position, please consider also in figure 7 (page 10) to change the wavenumber to an integer. Please according to it modify also lines 335 and 336.
  16. Please check the number of the figures. Figure 7 is reported twice on page 10 and page 12.
  17. In figure 8 on page 10, the entire inset could be added to the figure in the upper part.
  18. In line 380 you mention “ As shown in figure 9 in this complex sample the SERS spectrum exhibited several spectral signatures…” but I do not see the SERS spectrum in the such figure. It is necessary to check the number of the figures.
  19.  Consider adding at the end of the “Results and Discussion” a table including the main findings of your work (such as contaminants used, minimum concentration detected, and if was or was not possible to detect with the techniques used). This can help the reader to summarize your findings.

Reviewer 2 Report

This article evaluated Raman spectroscopy and SERS in detecting several chemicals and E. coli bacterium in water.

poorly written: sentences not clear, lots of run-on sentences

Require extensive editing by a technical editor proficient in English esp. the Introduction, Results & Discussion; need more info on Materials and Methods section

too wordy; simplify sentences

check ALL units format: examples         1mg/l = 1 mg/L;  1 µl = 1 µL

check word choices

spelling; run a spell-check

Check all chemical formulas:     sulfates (SO4-)

the Results and Discussion contain methods which should be under Materials and Methods

check general journal format: spacing, paragraph, sections

discuss instrument standardization prior to measurements

what are potential sources of variability in the measurement?

discuss: sample variability in different sample preparations; area where measurements were done

            “coffee ring” formation may not be uniform

specify sources of chemicals, purity, were chemicals obtained as standards? storage condition, were the chemicals freshly prepared? were the dissolved then dried samples scanned on the same day?

how were the ring diameter and thickness measured?

Microcystins can also

be bioaccumulated via the aquatic food chain and can be vectored

into terrestrial biota including humans.

line 25              word choice:  “well-being” is not normally used with the ecosystem         “…well-being of the ecosystem…”

            “Polluted waters represent a serious risk to human and ecological health and the well-being of the ecosystem, because…”

line 37              “…human health,  and animals (both domestic and wild) and , not worthy to mention plants and vegetables.

line 60  need comma:    “Until a few years ago, Raman…”

line 65  “…and to the use of…”

lines 66-67        “Further software development relies of on spectral identification software that have facilitated the research of on the spectral structures that characterize any analyte of interest.”

line 71  need  comma:   In SERS, the amplification…”

line 74 word choice, alternate words for “profit” : take advantage, benefit, )          “To profit from these…”

line 76 Thanks to this enhancement, the The SERS technique allows…”

lines 78  “…on the analyte considered.”

lines 79-82                  In this paper, we present the results of the application of Raman and SERS techniques for the detection of common aquatic anthropogenic chemical pollutants dissolved or suspended in water.”

lines 83-85                  The obtained results We showed that it was possible to detect these substances dissolved  in concentrations of a few ppm and in one case instance lower than the limit of concentration for the potability of water established by international regulatory bodies (need to specify).

line 90   identify this “GaAlAs “ first; gallium aluminum arsenide?

line 95   10X numerical aperture (N.A.)

lines 107 & 108   delete /

lines 106-118       simplify:

line 117               aluminum (Al) foil;    see line 182:    Al foil

The sample preparation was according to Xu et al. [16]. Briefly, a drop (2-5 µL) of dissolved substance in water on a microscope slide covered with aluminum foil at ambient temperature is evaporated forming a “coffee ring” [17] then scanned.

line 122 specify the meaning of g-NPS when first used

              gold nanoparticles?  gold nanoparticles (g-NPS)

line 123 lower case a in aluminum

lines 126-135       simplify, too much redundancy

) eutrophication of surface waters due to agricultural and urban runoff and sewage and industrial waste can lead to harmful blooms due to the availability of nutrients such as nitrogen and phosphrous. Massive and prolonged harmful blooms can also lead to hypoxia resulting in aquatic life die-off. Contamination of drinking water sources require multiple barrier approach in removing diverse contaminants either natural and/or anthropogenic which is often expensive

“1) eutrophication of surface waters, that is responsible for the excessive growth of plant organisms that occurs due to the presence in the aquatic ecosystem of too high doses of nutrients such as nitrogen, phosphorus or sulfur [18,19]. These come from natural or anthropogenic sources such as fertilizers, some types of detergents, wastewater of domestic or industrial origin, and the consequent degradation of the aquatic environment, which becomes asphyxiated, poor in oxygen and no longer suitable for the life of 131 fishes. While industrial pollutants include chemicals and clear hazardous materials, expanded populations in industrialized communities can also lead to higher bacterial contamination loads in basins and rivers representing a risk for public health with fecal pollution.”

line 137            remove extra period

line 136-146      use a) in line 139 and b) in line 143 since you already used numbers

lines 148-151    3) Microorganisms (viruses, bacteria, parasites, and fungi) from inadequately treated human communities’ wastewater inadequate treatment that can be responsible for intimate can increase the risk of human exposure. Waterborne pathogens can cause severe diseases in humans [22], especially if we consider breakdown of sewage treatment system breaks down, during sewage overflows and unintentional contamination of internal potable waters and food supplies.

lines 172, 174, 202        lower case a and n in ammonium nitrate

lines 177-179              The contamination of water by ammonium nitrate was simulated preparing solutions at known concentration of NH4NO3, both in de-ionized (DI) water (18.2 M ∙ cm) both in drinking water (drawn from the southern Rome area) and finally adding an aliquot of g-NPS with concentration of about 5 ∙ 10-4 mg / ml.

lines 180, 181, 197, 198, etc…   check journal format in writing: 1mg/l = 1 mg/L;  1 µl = 1 µL

5 ∙ 10-4 mg / ml (5 x 10-4 mg/mL); 405 mg / L = 405 mg/mL; 17.2 mg / L = 172. mg/L

            Check ALL units

lines 192-194    In the case of drinking water, the characteristic NO3- peak at 1046 cm-1 was still detected but at  with lower sensitivity, in a solution with a minimum concentration of 20 mg/L NH4NO3, but still within (specify the law here) the Italian law and WHO limits for nitrates in water. The corresponding SERS spectrum is reported in Figure 3. As previously reported already observed in literature, [29] it is supposed that the concurrent presence of other…”

In drinking water, the characteristic NO3- peak at 1046 cm-1 was still detected but at lower sensitivity in a solution with a minimum concentration of 20 mg/L NH4NO3 but still within (specify the law here) the Italian law and WHO limits for nitrates in water. The corresponding SERS spectrum is reported in Figure 3. As previously reported [29], the presence of other…”

line 198            ”… SERS "hot spots", i.e. the…”   i.e.,

lines 207-208    The presence of phosphates in internal water is linked to the entry into the water body of

(also lines 244 & 392     inner waters”; not sure what you mean by “inner,” “internal,” or “inland” water.  Do you mean surface water/freshwater/sources of drinking water?

Is this what you mean in lines 207-208?

The following are sources of phosphates that feed into bodies of water:

line 212             delete period

line 217            correct:             orthophosphate (PO43–)

lines 217-224  fix this; is this a whole paragraph?  very sloppy

line 222           spelling:          eutrofication

line 225           indent for paragraph

lines 228, 239, 272, etc…       check format for in text Figure #

line 251           “…delay degenerative processes.” do you mean degradation processes?

Section 3.3 Sulphites     “yeasts present on the skin of the berries” repeated

line 279            SO3 clarify; in solution it has a charge of -2

line 284             suggested by according to the WHO in the Guidelines for Drinking Water Quality

line 288            whitin the suggested within the WHO guideline limit.

line 389            3.5 Escherichia coli

lines 394-397       poorly written, revise:       The presence of pathogenic microorganism in groundwater is also important because, due to the recent prolonged period of drought and the pressing demand of water for crop there is a general intention of using for this purpouse wastewater for irrigation (FAO estimates that the 70%of the water consumption is destined to agriculture).”

spelling purpose

Define FAO

lines 402-405    revise, run-on sentence; sentence convoluted; break the sentence so it makes sense

            check spelling:  anthropic” action pollution; not sure what you mean

lines 411-413       how was log phase determined?  how did you determine the cell count and viability?

              why were the bacteria washed and resuspended in Milli-Q® deionized water?

line 429 not sure what you mean by “did not give significant results”

I really do not have the time to edit this manuscript.

Round 2

Reviewer 1 Report

Dear Authors,

Thank you for taking my comments into account.

Kind Regards.

Author Response

Thanks to the reviewer for his comments and observations: They were useful to improve the quality of the manuscript

Reviewer 2 Report

A “clean” version would have been easier to review; unfortunately, my system does not have this capability.

 Much improved! but can still be improved

check to make sure chemicals are in lower case (e.g., glyphosphate, methanol, etc.)

Introduction

Combine paragraphs 1 & 2

Can include a brief info about the contaminants selected/detected in the study; just move some of the info you deleted here

line 141 “…and can be easier easily measured, also in an in-situ campaign setting.

line 198                 “…generating anomalous algal flowers (?; do you mean algal bloom?.”

lines 229-230      “…both in deionized water (18.2 MW ∙ cm) both in DI and drinking water (drawn from the southern…”

line 233                 Phosphates, their The presence of phosphates in inland waters comes…”

line 244                 “Sulfites, they are preservatives used for food conservation, to…”

line 263                 “ Polycyclic aromatic hydrocarbons (PAHs),  are present…”

line 277                 delete comma:  “… and, finally…”

line 284                 check format and be consistent: (2.5∙10-3 mg)      line 294    7.9∙10-4 M

                                line 279    (1.12×10-3 M)

line 297 Escherichia coli (EC)        italicize

                used an indicator of human pollution

                “…environment due to anthropic activities,…”

lines 323-324      “Although G glyphosate is one of the major a widely used herbicide s employed, recently the dangerousness hazard and toxicity of glyphosate and AMPA has been demonstrate for human health [29-31,49] and the environment”

line 326                 From In 2015, the WHO ’s International Agency

lines 327-329      “Even if the toxicity of AMPA was reported to be less than or comparable  to glyphosate or comparable [51], due to its protracted persistence in the environment, several studies established that environmental AMPA harmfulness  toxicity was higher than that of glyphosate [52,53].”

is this what you mean?

line 341                 “All the solutions/suspensions have been were prepared and measured on the same day

line 358                 “The nitrate ion was in DI water at 1 mg/L.”

line 372 “…L NH4NO3 ,  which is  still within the…”

line 476                 spelling of within              “… concentrations within the suggested WHO limit

line 510                 “… because it is persistent, can bioaccumulate ing and toxic for to aquatic organisms.”

lines 523-526      sentence fragmented:   “As shown in Figure 8, the spectral signatures of anthracene in a solution were evidently detected at about 394 cm-1 , 751 cm-1 , and 400 cm-1 in agreement with literature data [48]63]. from a solution

line 555                 word choice & spelling   “… that the whole entire substance dissolved…”

line 557                 comma:                “With a laser spot of 40 mm diameter, the

lines 583-585      “As shown in Figure 10, in this complex mixture sample, the SERS spectrum exhibited several spectral signatures ascribable to single component s of the mixture, which is in good agreement with the results reported in literature [64].”

lines 638-639      need a comma: “To overcome this problem, we applied a preconcentration step recommended by

 line 646               ? (specify)           “… assignments according to Kusíc et al. [68]

line 692                 spelling and lower case g              “The Raman spectrum of glyphosate in crystalline form

line 710                 “… .found correspondent.” what do you mean “correspondent”?

                                similar?

line 727                 “… SERS analysis of samples of in DI and

line 732                 “… substances that can hardly be that are difficult to measure measured

Additional comments/suggestions

check setting a paragraph; indent?

need to go over the proper use of punctuation marks

line 22                  lower case “c” & italicize Escherichia coli

line 71                  “… The result is an intense elec-

lines 75-79           usually, a paragraph contains more than a sentence

                              can combine these two paragraphs

lines 98-100         “Measurements were performed by using the 40X objective, representing, the right compromise between a relatively large scanned area, that limits the fluctuations of the Raman-SERS signal in different points of the sample and a good collection efficiency, increasing SNR.”

lines 113-116       it may be good to include a scan at different points in a ring to show variability

line 123               “… Al-covered microscope glass slide.”

line 124 “We used common drinking water but also  and de-ionized (DI) water as solvent, to high-

lines 131-132       grammar              “Massive and prolonged algal blooms leads to hypoxia, resulting in aquatic life die-off or large pollution of water by microcystins.”

Methods and Materials has info on the reason the chemicals and microorganism were selected which should be in the Introduction

line 237                “… warm blooded animals

line 273               “… glyphosate and AMPA, are not normally mandatory in routine sampling monitoring

lines 302-303     I suggest specifying the “Italian law”

sentence fragmented:       “… minimum concentration of 20 mg/L NH4NO3, which is still within the limits for nitrate ions dissolved in water of  in the Italian law [34] and WHO for nitrate ions dissolved in water [35].”

lines 322-324 this sentence is awkward; revise:   “Specifically, the PO43- peak at 915 cm-1  as revealed with evidence for concentrations not less than (is this what you mean???) ≥ 1 mg/mL, increasing with the acquisition time up to 120 s and the laser power up to 300 mW. The resulting Raman spectrum is shown in figure 5.

the next sentences up to line 330 can be improved, too

line 395                “… in M methanol (MEOH), containing these compounds

lines 324, 400, 442                         capitalize or lower case F in Figure X in text? check journal format

line 441               lower case g in glyphosate

line 472               “In this study, we report the results of a Raman/SERS analysis of samples of DI and drinking water contaminated spiked with substances chemicals and microorganisms representative

lines 474-475     “The coffee-ring methodology was used was that of the coffee-ring and was chosen to

Conclusion

check paragraph setting; indent?

Author Response

Dear referee. We thank you for your comments and observations aiming to improve theoverall quality of the manuscript. Attached you find the .docx file containing answers to them. 
